# Plasma Atrial Natriuretic Peptide Predicts Oxidized Low-Density Lipoprotein Levels in Type 2 Diabetes Mellitus Patients Independent of Circulating Adipokine and Cytokine

**DOI:** 10.3390/ijms26051859

**Published:** 2025-02-21

**Authors:** Ahmed Bakillah, Maram Al Subaiee, Ayman Farouk Soliman, Khamis Khamees Obeid, Shahinaz Faisal Bashir, Arwa Al Hussaini, Mohammad Al Arab, Abeer Al Otaibi, Sindiyan Al Shaikh Mubarak, Ali Ahmed Al Qarni

**Affiliations:** 1King Abdullah International Medical Research Center (KAIMRC), Eastern Region, Al Mubarraz 36428, Saudi Arabia; bashirsha@kaimrc.edu.sa (S.F.B.); alhussainiar@kaimrc.edu.sa (A.A.H.); alarabmo84@gmail.com (M.A.A.); otaibiabe@kaimrc.edu.sa (A.A.O.); alshaikhmubaraks@kaimrc.edu.sa (S.A.S.M.); qarniaa@mngha.med.sa (A.A.A.Q.); 2Division of Biomedical Research Core Facility, King Saud bin Abdulaziz University for Health Sciences (KSAU-HS), Al Mubarraz 36428, Saudi Arabia; 3Ministry of National Guard-Health Affairs (MNGHA), King Abdulaziz Hospital, Al Mubarraz 36428, Saudi Arabia; alsubaieema@mngha.med.sa (M.A.S.); solimana@mngha.med.sa (A.F.S.); khamisko@mngha.med.sa (K.K.O.)

**Keywords:** type 2 diabetes, dyslipidemia, insulin resistance, obesity, adipokines, cytokines, ANP, ox-LDL, vascular complications, coronary artery disease, endothelial dysfunction, inflammation

## Abstract

Atrial natriuretic peptide (ANP) and oxidized low-density lipoprotein (ox-LDL) play essential roles in the development and progression of vascular complications associated with type 2 diabetes mellitus (T2DM), and both are independently linked to cardiovascular diseases (CVD). However, the relationship between ANP and ox-LDL in patients with T2DM remains unclear as previous studies have primarily focused on circulating levels in various diseases. This study investigated the relationship between ANP and ox-LDL levels in obese individuals with T2DM. The cohort included 57 patients with T2DM (mean age 61.14 ± 9.99 years; HbA1c 8.66 ± 1.60%; BMI 35.15 ± 6.65 kg/m^2^). Notably, 95% of the patients had hypertension, 82% had dyslipidemia, 59% had an estimated glomerular filtration rate (eGFR) < 60 mL/min/1.73 m^2^, 14% had coronary artery disease (CAD), and 5% had a history of stroke. Plasma concentrations of ANP and ox-LDL were measured using ELISA. Adipokines and cytokines levels were measured using the multiplex^®^ MAP Human Adipokine Magnetic Beads Spearman’s correlation analysis which revealed a negative correlation between ANP and ox-LDL (r = −0.446, *p* = 0.001) as well as with the ox-LDL/apoB ratio (r = −0.423, *p* = 0.001) and ox-LDL/LDLc ratio (r = −0.307, *p* = 0.038). Multivariable regression analysis indicated that ANP was independently associated with ox-LDL (β = −115.736, *p* = 0.005). Stepwise linear regression further identified TNFα, leptin, and adiponectin as the strongest predictors influencing the relationship between ANP and ox-LDL levels (β = −64.664, *p* = 0.0311, and r^2^ = 0.546 for the model). However, these factors did not significantly mediate this association. This study emphasizes the need for further exploration of the complex interaction between ANP and ox-LDL in larger patient populations. This could provide valuable insights into potential therapeutic approaches for managing vascular complications in obese individuals with T2DM.

## 1. Introduction

Obesity and diabetes are significant global health challenges that are closely associated with elevated risk of cardiovascular diseases (CVD) [1,2]. These two conditions contribute to various metabolic and vascular changes, including dyslipidemia, endothelial dysfunction, chronic low-grade inflammation, and increased oxidative stress. Insulin resistance, a hallmark of type 2 diabetes (T2DM), is associated with increased oxidative stress and the production of pro-inflammatory substances, which further exacerbate endothelial dysfunction [3]. The accumulation of excess adipose tissue, particularly visceral fat, triggers the release of pro-inflammatory adipokines, worsening the risk of cardiovascular events in obese individuals with diabetes [4].

Atrial natriuretic peptide (ANP) and oxidized low-density lipoprotein (ox-LDL) are emerging biomarkers that play essential roles in the pathophysiology of obesity-related T2DM. ANP is a peptide hormone mainly secreted by the atria of the heart in response to increased blood volume and pressure. It offers several cardiovascular benefits, including promoting natriuresis, diuresis, and vasodilation [5]. Additionally, ANP also possesses anti-inflammatory and antioxidant properties, which may help protect against cardiovascular complications. However, in conditions such as obesity and diabetes, the regulation and effectiveness of ANP may be impaired, contributing to an increased risk of heart failure and atherosclerosis [6].

Ox-LDL is a modified form of LDL cholesterol that significantly contributes to atherosclerosis [7,8]. It promotes endothelial dysfunction, inflammation, and foam cell formation, which triggers the formation of atherosclerotic plaque within arterial walls [9,10]. In obesity and diabetes, oxidative stress is often elevated, leading to the increased oxidation of LDL particles. These changes exacerbate further vascular injury and heighten the risk of CVD in obese diabetic patients [11].

The current body of research supports the idea that both ANP and ox-LDL are crucial factors in the development of CVD in obese and diabetic patients. While ANP seems to have protective effects against oxidative stress and inflammation, its effectiveness is often reduced in the presence of obesity and diabetes [12]. Conversely, elevated levels of ox-LDL worsen oxidative stress and inflammation, driving cardiovascular complications [13].

Research shows that ANP levels are often lower in obese individuals compared to non-obese individuals, and this reduction is even more pronounced in diabetic patients [14,15,16]. Increased adiposity was inversely related to ANP levels, suggesting that excess body fat interferes with ANP production or release [17]. Lower ANP levels have consistently been observed in individuals with insulin resistance, implying that insulin resistance may directly affect ANP secretion or function [18,19]. Although ANP levels are lower, ANP may provide some of its cardiovascular protective effects in individuals with obesity and diabetes to reduce vascular stiffness and inflammation by regulating pro-inflammatory cytokines [20]. Therefore, reduced ANP function may limit its effectiveness in obese and diabetic individuals, resulting in higher cardiovascular morbidity in this population.

Studies have indicated that ANP treatment reduced oxidative stress markers and lowered ox-LDL levels in atherosclerosis animal models, suggesting that ANP may counteract the harmful effects of LDL oxidation through its antioxidant properties [21,22,23]. Additionally, ANP could suppress the expression of pro-inflammatory cytokines, mitigating the inflammatory responses triggered by ox-LDL. Studies also indicated that elevated ox-LDL levels may impair ANP function. High ox-LDL concentrations may reduce the expression of ANP receptors in vascular smooth muscle cells [24], leading to a diminished response to ANP and contributing to vascular dysfunction, thus increasing cardiovascular risk in obese diabetic patients.

Hormones such as adipokines and cytokines play a critical role in the pathophysiology of obesity and diabetes by influencing inflammation, insulin sensitivity, fat metabolism, and glucose regulation [25]. Adipokines like leptin and resistin may lower ANP levels, while adiponectin may enhance its natriuretic effects [20,26,27]. Similarly, cytokines like TNF-α and IL-6 can suppress ANP secretion, impairing its natriuretic and vasodilatory effects and exacerbating fluid retention and blood pressure issues in diabetic patients [26]. Furthermore, Adipokines can enhance the oxidation of LDL, whereas adiponectin helps reduce ox-LDL levels. Pro-inflammatory cytokines, including TNF-α, IL-6, and IL-1β contribute to oxidative stress, leading to LDL oxidation and an increased risk of atherosclerosis [28,29,30,31].

The interplay between ANP, ox-LDL, and circulating hormones like adipokines and cytokines creates a complex network of interactions that can significantly affect the relationship between ANP and ox-LDL. Therefore, we aimed to examine the association between ANP and ox-LDL levels in obese diabetic patients to clarify potential mechanisms linking these two factors.

## 2. Results

### 2.1. Baseline Characteristics of the Study Subjects with T2DM

The patient characteristics of the study subjects with T2DM are detailed in Table 1. The mean patient age was 60.96 ± 9.99 years (female 44%). Ninety-five percent (95%) had hypertension, 79% had dyslipidemia, and 14% had CAD. Mean HbA1c levels were 8.66% ± 1.60%. Most subjects were overweight (Mean BMI = 35.15 ± 6.65 kg/m^2^) with a mean eGRF of 58.04 ± 26.50 mL/min/1.73 m^2^. All subjects were Saudi citizens with 82% and 58% having a family history of diabetes and cholesterol, respectively. The mean values of plasma ANP and ox-LDL as determined by ELISA were 0.47 ± 0.22 pg/mL and 247.97 ± 67.50 ng/mL, respectively.

### 2.2. Quantification of and Distribution of Plasma ANP and ox-LDL Among T2DM Patients

Plasma ANP values among diabetic patients showed a right-skewed distribution with a median value of 0.43 [0.32, 0.58] pg/mL. In contrast, values of ox-LDL were normally distributed in this patients group, with a median value of 237.50 [198.78, 288.45] ng/mL (Figure 1A,B). Plasma ANP was inversely correlated with ox-LDL ratio and ox-LDL/apoB ratio; Figure 1C–E.

### 2.3. Correlations Between Plasma ANP, ox-LDL Levels, and Clinical Parameters in T2DM Patients

Spearman’s correlation analysis revealed that plasma ANP negatively correlated with Ox-LDL (r= −0.446, *p* = 0.001; Table 2), Ox-LDL/ApoB (r= −0.423, *p* = 0.001; Table 2), Ox-LDL/LDL-c (r= −0.307, *p* = 0.038; Table 2), and positively correlated with BMI (r = 0.297, *p* = 0.025; Table 2). There was no significant correlation of ANP with other parameters such as age, gender, hypertension, HbA1c, creatinine, total cholesterol, LDL-c, HDL-c, and triglycerides (Table 2). Among all selected adipokines and cytokines, ANP was positively correlated with adiponectin (r= 0.288, *p*= 0.030) and negatively correlated with IL6 (r = −0.305, *p* = 0.021; Table 3). In contrast, ox-LDL was positively correlated with IL6 (r = 0.340, *p* = 0.010), leptin (r = 0.353, *p* = 0.007), and TNFα (r = 0.530, *p* = 0.001), and negatively correlated with adiponectin (r = −0.465, *p* = 0.001); Table 3.

### 2.4. Univariate Regression Analysis for the Association of Plasma ANP with ox-LDL in T2DM Patients

A univariate analysis investigated the association between plasma ANP and Ox-LDL levels. The results indicated a negative association between circulating ANP and Ox-LDL levels (β = −117.23, *p* = 0.004; Table 4). Additionally, the Ox-LDL/ApoB ratio was significantly associated with ANP, while the Ox-LDL/LDL-c ratio was not (β = −1.874, *p* = 0.016; Table 4). This indicates that the ox-LDL/apoB ratio, which shows the extent of the oxidation of the LDL particles and the ox-LDL/LDL-c ratio, which represents the level of LDL cholesterol oxidation, may provide different insights regarding ox-LDL interaction with ANP.

### 2.5. Multivariate Regression Analysis for the Association Between Plasma ANP and ox-LDL in T2DM Patients

We also explored the potential role of other variables that may explain the relationship between ANP and ox-LDL concentrations. A multi-regression analysis was performed after adjusting the regression models for various independent variables such as age, gender, HbA1c, total cholesterol, LDL-c, HDL-c, triglycerides, Hs-CRP, and BMI. The analysis indicated that plasma ANP was the strongest predictor of ox-LDL levels, independent of age, gender, and HbA1c (Models M1 and M2; Table 5A). However, this association disappeared when other independent variables, such as total cholesterol, LDL-c, HDL-c, triglycerides, Hs-CRP, and BMI, were included in the regression model (Model M3; Table 5A). Additionally, ANP was significantly associated with the ox-LDL/ApoB ratio but not with the ox-LDL/LDL-c ratio (Model M1; Table 5B,C).

### 2.6. Stepwise Regression Analysis for the Identification of the Best Predictors for the Association with ox-LDL Among Circulating Adipokines and Cytokines in T2DM Patients

To further evaluate the influence of various circulating adipokines and cytokines on the relationship between ANP and Ox-LDL, we first conducted a multi-regression analysis that included all selected adipokines and cytokines in the model. The results indicated that ANP significantly predicted ox-LDL levels (β = −72.823, *p* = 0.030), explaining 61% of the variance in the constructed model (Table 6A). Among all the variables examined, leptin, TNFα, and adiponectin were identified as significant confounding factors in the regression model (*p* = 0.034, *p* = 0.038, and *p* = 0.001, respectively; Table 6A). Next, we performed a stepwise linear regression to identify optimal subsets of potential predictors with estimated coefficients that best predict the relationship between ANP and Ox-LDL (Table 6B). Independent variables from the selected adipokines and cytokines were added to the regression models along with ANP. The results showed that TNFα, leptin, and adiponectin had the most significant impacts when included in the predictive model (Model M4; Table 6B), explaining 54% of the variability in the relationship between ANP and Ox-LDL.

### 2.7. Regression Analysis for Potential Mediating Variable Effect on the Relationship Between ANP and ox-LDL in T2DM Patients

To further investigate whether the association between the ANP (independent variable) and Ox-LDL (dependent variable) could be indirectly explained by any of the selected adipokines and cytokines (mediator), we conducted sequential regression analyses using PROCESS macro V4.2 by Andrew F. Hayes (model 4). The path model simultaneously tested three effects for each potential mediator (Table 7): (i) the effect of the independent variable (ANP) on the mediator (indirect effect path a); (ii) the effect of the mediator on the dependent variable (Ox-LDL), known as the indirect effect path b; (iii) the mediation effect (a*b), which represents the reduction in the relationship between ANP and Ox-LDL (total effect path c) when the mediator is included in the model (direct effect path c’). Among all the variables tested, TNFα and adiponectin exhibited partial moderation of the relationship between ANP and Ox-LDL, with mediation effects of 24% and 14%, respectively. However, these observed mediating effects were not statistically significant (Table 7).

## 3. Discussion

Obesity and T2DM are well-established risk factors for a large number of cardiovascular complications, including endothelial dysfunction and atherosclerosis [3,32]. Although ANP and ox-LDL have been extensively studied individually in various disease conditions, to our knowledge, there is no information on the relationship between their levels in obese individuals with T2DM [9,33,34,35,36,37]. Targeting therapies aiming to reduce ox-LDL levels or boost ANP signaling may present effective methods for enhancing vascular health and lowering cardiovascular risk in obese individuals with T2DM. Therefore, our primary goal was to investigate the relationship between circulating plasma ANP and ox-LDL levels in the context of the metabolic and hormonal fluctuations linked to obesity and T2DM.

In obese individuals with T2DM, the combined effects of hyperglycemia, dyslipidemia, and inflammation further elevate ox-LDL levels, exacerbating endothelial dysfunction and increasing the risk of cardiovascular complications [7]. Our study found that patients with T2DM had elevated circulating ox-LDL levels and a slight, though not statistically significant, decrease in plasma ANP levels compared to healthy controls. The increase in ox-LDL levels was consistent with previously published studies [38,39,40]. In contrast, conflicting results concerning ANP levels have been reported [12,41,42,43].

Recent studies suggest that ANP may directly affect lipid metabolism by influencing lipogenesis, adipogenesis, and fat mobilization from adipose tissue, potentially altering the lipid profile in individuals with obesity or diabetes [44,45,46,47,48,49]. However, the exact nature of the relationship between ANP and lipid metabolism remains unexplored. Previous studies have reported a significant association between higher ANP levels and a favorable lipid profile [50]. This study found no significant correlations between ANP and ox-LDL with traditional lipid parameters such as total cholesterol, LDL-c, HDL-c, and triglycerides, nor with HbA1c and BMI. The lack of associations between ANP and ox-LDL with lipid parameters may be attributed to disturbances in the lipid profile, which could lead to elevated levels of ox-LDL and increased oxidative stress. ANP levels were frequently altered in T2DM, with some studies reporting elevated circulating levels [14,41]. These elevated levels in circulating ANP levels are believed to be a compensatory response to counterbalance the increased blood volume and altered hemodynamics resulting from obesity and insulin resistance. Nevertheless, despite this compensatory mechanism, cardiovascular effects of ANP may be diminished in T2DM due to receptor desensitization, impaired ANP signaling, and/or changes in renal function that could affect ANP clearance [51,52,53,54].

Adipokines and cytokines play a significant role in regulating adipose tissue inflammation and metabolic dysfunction in diabetic patients, which can influence levels of both ANP and ox-LDL. Increased pro-inflammatory cytokines such as TNF-α and IL-6, along with adiponectin, have been shown to contribute to endothelial dysfunction, oxidative stress, and insulin resistance [55,56]. Simultaneously, the upregulation of inflammatory markers promotes the oxidation of LDL particles, resulting in elevated ox-LDL levels [57]. This cascade of events exacerbates vascular damage, further increasing the risk of cardiovascular complications in diabetic individuals [13,58,59,60].

In this study, we observed that ANP exhibited a positive correlation with adiponectin and a negative correlation with IL-6. In contrast, ox-LDL had a negative correlation with adiponectin and a positive correlation with IL-6, leptin, and TNFα. These findings align with previous studies, which suggest that variations in the levels of adipokines such as adiponectin and leptin, along with inflammatory cytokines like IL-6 and TNF-α, can influence both ANP and ox-LDL levels. These results further support the notion that the imbalance of adipokines and inflammatory markers may significantly influence cardiovascular risk factors in individuals with diabetes [41,60,61,62,63,64]. Notably, adiponectin, generally considered protective, negatively correlates with ox-LDL, whereas IL-6 can elevate ox-LDL, thereby contributing to oxidative stress and vascular complications [60,61]. Understanding the interplay between these factors is crucial, as alterations in adipokine and cytokine levels can significantly influence ANP and ox-LDL levels, potentially impacting the development of cardiovascular complications in obese individuals with T2DM. We hypothesize that adipokines and inflammatory cytokines may mediate the relationship between ANP and ox-LDL, potentially by playing a role in vascular dysfunction and increased cardiovascular risk commonly observed in obese individuals with T2DM.

The multiple regression analyses in this study revealed that, among all the selected adipokines and cytokines, only TNFα and adiponectin were identified as potential confounding factors that significantly contributed to the relationship between ANP and ox-LDL. However, their effects did not result in any apparent mediating effect. The lack of a mediating effect could be attributed to several reasons, including the nature of the relationship, limitations in the study design and sample size, or the complexity of the biological mechanisms involved in the ANP signaling and LDL oxidation. Additionally, the interaction between ANP and ox-LDL may involve a complex network of interactions where mediating factors might only play a significant role under specific conditions or in certain populations. Another plausible explanation is that other unmeasured confounders may be implicated in the relationship between ANP and ox-LDL, potentially masking the anticipated mediating effect of the selected variables. Furthermore, the underlying mechanisms may involve factors that may not have been captured in the analysis.

The relationship between ANP and ox-LDL is crucial for cardiovascular health, particularly in inflammation, lipid metabolism, and vascular function. This relationship becomes more complex in obese individuals with T2DM where metabolic dysregulation can increase cardiovascular risks. In T2DM, elevated oxidative stress leads to the oxidation of LDL into ox-LDL, contributing to cardiovascular complications by promoting endothelial dysfunction, inflammation, and atherosclerotic plaque formation [65]. Although ANP levels are often elevated in the early stages of T2DM due to compensatory mechanisms related to fluid and blood pressure regulation, this protective function is frequently compromised by insulin resistance, oxidative stress, and other metabolic changes, diminishing its effectiveness against cardiovascular complications [42]. As T2DM progresses to later stages, ANP levels may decline, worsening vascular and renal dysfunction. Research indicates that ANP can mitigate these adverse effects through various mechanisms. Firstly, ANP reduces oxidative stress by activating antioxidant pathways, which helps limit ox-LDL accumulation. This reduction in oxidative stress indirectly inhibits the inflammatory processes triggered by ox-LDL. Additionally, ANP has anti-inflammatory effects by modulating inflammatory cytokine production, decreasing inflammatory cell activation, and promoting vascular health. ANP also regulates lipid metabolism by improving lipid profiles and reducing ox-LDL concentrations [66]. Furthermore, ANP enhances endothelial function by promoting vasodilation and protecting against damage caused by oxidized lipoproteins, contributing to improved vascular tone and a reduced risk of atherosclerosis.

Several possible mechanisms could explain the association between ANP and ox-LDL. In T2DM, elevated oxidative stress increases reactive oxygen species (ROS) in individuals with obesity, which may lead to enhanced oxidation and, consequently, higher levels of ox-LDL [67]. This, in turn, could trigger an inflammatory response that impacts the cardiovascular system, prompting the secretion of ANP as a compensatory mechanism to address vascular dysfunction [68]. Ox-LDL has been shown to impair nitric oxide (NO)-mediated vasodilation, which could disrupt the interaction between ANP and NO and further complicate vascular regulation [69,70,71]. Additionally, the inflammatory effects of ox-LDL may stimulate ANP release as part of a counter-regulatory response to maintain vascular tone and fluid balance. However, in the context of chronic inflammation seen in T2DM and obesity, the protective effects of ANP may be diminished. While ANP has anti-inflammatory properties, its effectiveness may be compromised by the pro-inflammatory environment caused by ox-LDL. Finally, ANP’s potential role in lipid metabolism could explain its relationship with ox-LDL. Some studies suggested that ANP may influence lipid metabolism by modulating adipokine release from adipose tissue or altering lipoprotein particle composition [72]. Since ox-LDL is closely associated with dyslipidemia and atherosclerosis, ANP may help mitigate the effects of lipid imbalances in obese individuals with T2DM, potentially affecting the production or clearance of ox-LDL.

One of the main limitations of this study is its relatively small sample size and the lack of a healthy control group, which may have reduced the statistical power and generalizability of the findings. A larger population of T2DM subjects and a well-matched control group would confirm the observed association between ANP and ox-LDL and provide more robust conclusions. Additionally, the study was conducted at a single center, which may introduce biases related to patient selection and limit the applicability of the results to broader populations. Another potential limitation is the inability to account for all possible confounding factors, as there may be undetected variables that could influence the outcomes. Finally, the study focused on obese individuals with T2DM who have specific characteristics. Therefore, our findings may not fully apply to other populations or different stages of the disease. Further studies with larger, more diverse populations and a broader range of potential confounders are needed to validate these findings.

## 4. Materials and Methods

### 4.1. Study Population and Protocol

This is a retrospective study that includes 57 patients with T2DM and 13 healthy individuals recruited from clinics at King Abdulaziz Hospital (KAH), Ministry of National Guard-Health Affairs at Al-Ahsa, Kingdom of Saudi Arabia between January 2020 and April 2021. The study protocol was approved by the Institutional Review Board of the Ministry of National Guard, Health Affairs (IRB protocol# IRBC/1972/18), and written informed consent was obtained from each participant. Participants were excluded from the study if, at baseline, patients met one or more of the following criteria: patients were receiving chronic renal replacement therapy (hemodialysis, peritoneal dialysis, or transplantation), had a history of active malignancy (except those with basal cell carcinoma) within the last five years (prostatic cancer within the past two years), systemic lupus erythematosus and other autoimmune diseases that may affect kidney function, history of type 1 diabetes, acute infection or fever, pregnancy, chronic viral hepatitis or HIV infection, and current unstable cardiac disease. The following standard methods and definitions were adopted: Diabetes: subjects with a history of T2DM on medication, HbA1c ≥ 6.5%, or fasting glucose ≥ 126 mg/dL (≥7 mmol/L). The family history of T2DM was defined as any first-degree relative diagnosed with T2DM. Dyslipidemia: subjects with a history of dyslipidemia on medication or fasting lipid profile with total cholesterol > 200 mg/dL or LDL >70 mg/dL. Hypertension: subjects with systolic blood pressure ≥ 140 mmHg or diastolic blood pressure ≥ 90 mmHg and under antihypertensive medication use. CKD: subjects with eGFR < 90 mL/min, using a modification of diet in renal disease (MDRD) equation or proteinuria (≥2+ on urine dipstick).

### 4.2. Measurement of Plasma Levels of ANP, Ox-LDL, Adipokines, and Cytokines in T2DM Patients

Fasting blood samples were collected in the morning after a minimum of 12 h fasting into EDTA-containing tubes and centrifuged at 4 °C at 3000 rpm for 10 min to separate the plasma for biochemical tests. Samples from T2DM patients and healthy subjects were aliquoted and stored at −80 °C until further analysis. Patient medical history, demographics, and laboratory parameters were collected from the electronic medical record in the BEST Care database. Plasma concentrations of ANP and ox-LDL were determined by ELISA using commercially available kits according to the manufacturer’s protocols (catalog# E-ELH0532 and catalog# E-EL-H6021, respectively, from Elabscience Biotechnology Inc., Houston, TX, USA). Adipokines and cytokines levels were measured using the multiplex^®^ MAP Human Adipokine Magnetic Bead Panel 2 (catalog# HADK2MAG-61K; EMD Millipore-Sigma; Burlington, MA, USA). Human adiponectin/ACRP30 concentrations were determined using a commercially available ELISA kit (catalog# E-EL-H6122; Elabscience Biotechnology Inc., Houston, TX, USA). According to the manufacturer, the sensitivity of the ANP kit is 4.69 pg/mL, with a detection range of 7.81 to 500 pg/mL. In contrast, the ox-LDL kit has a sensitivity of 37.5 pg/mL, with a detection range of 62.5 to 4000 pg/mL. These two kits specifically identify human ANP and ox-LDL in samples such as serum, plasma, and other biological fluids, demonstrating no significant cross-reactivity. All determinations were performed in duplicates.

### 4.3. Statistical Analysis

Statistical analyses were conducted using SPSS software version 30.0 (IBM Corp., Chicago, IL, USA). The normal distribution of the data was assessed using the Kolmogorov–Smirnov test. Continuous variables following a normal distribution were presented as means ± SD, while categorical variables were reported as frequencies and percentages. To evaluate the relationships between plasma ANP, ox-LDL levels, and other variables, we used the non-parametric Spearman’s correlation test. We performed a multiple regression analysis to explore the association between ANP and ox-LDL using adjusted-based models for covariates such as age, sex, HbA1c, creatinine, BMI, total cholesterol, LDL-c, HDL-c, triglycerides, and other factors, including adipokines and cytokines. Additionally, we conducted a stepwise analysis to identify the optimal subset of predictor variables related to the association between ANP and ox-LDL. We reported the outcome analysis as adjusted β-coefficients and 95% confidence intervals. For statistical significance, two-sided tests with *p*-values < 0.05 were considered statistically significant. To evaluate how much the measured effect of the independent variable (ANP) on the dependent variable (ox-LDL) could be attributed to a potential mediator variable (adipokines or cytokines), we assessed the mediation effect in sequential regression analyses using PROCESS macro V4.2 by Andrew F. Hayes (model 4). We applied bootstrapping methods to calculate 95% confidence intervals of coefficients for the total, indirect, and direct effects coefficients. Partial mediation was detected when the mediating variable significantly reduced the association between ANP and ox-LDL. The mediation effect percentage was defined as the ratio of the indirect effect to the total effect.

## 5. Conclusions

Collectively, both ANP and ox-LDL are important in modulating CVD, and their interaction may further promote vascular dysfunction and elevate cardiovascular risk among obese patients with T2DM. Furthermore, studies are needed to clarify and better elucidate the mechanisms linking circulating ANP and ox-LDL and to establish the causal relationship and clinical relevance of these markers in the context of obesity and T2DM treatment. Changes in plasma levels of ox-LDL and ANP might serve as potential biomarkers for assessing cardiovascular risk and guiding treatment strategies in obese patients with T2DM.

## Figures and Tables

**Figure 1 ijms-26-01859-f001:**
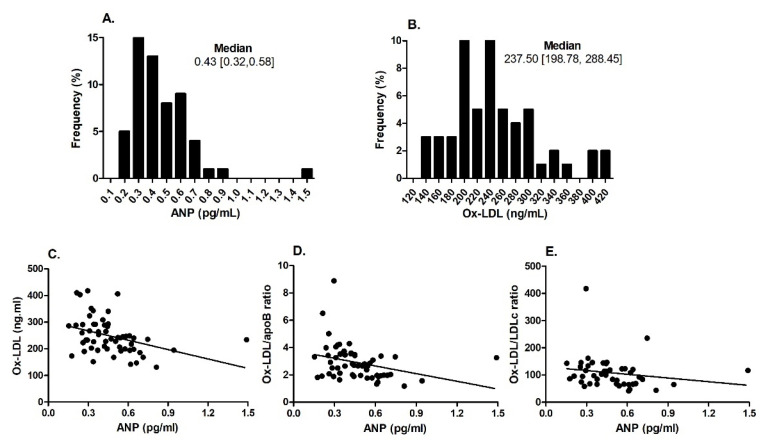
The circulating levels of ANP and ox-LDL in T2DM patients were measured using ELISA. The frequency distribution of ANP and ox-LDL was calculated for the diabetic subjects (Panels (**A**,**B**)). Correlations between plasma ANP and ox-LDL, as well as ox-LDL/ApoB and ox-LDL/LDC ratios are displayed in Panels (**C**–**E**), respectively.

**Table 1 ijms-26-01859-t001:** Baseline characteristics of 57 patients with T2DM.

Baseline Characteristic	Value
Age (years) *	60.96 ± 9.99
BMI (kg/m^2^) *	35.15 ± 6.65
Systolic BP (mmHg)	145.00 (123.50, 153.50)
Diastolic BP (mmHg)	71.00 (61.50, 81.00)
HbA1c (%) *	8.66 ± 1.60
Fasting Glucose (mg/dL)	8.00 (6.67, 14.27)
Hs-CRP (mg/L)	7.60 (3.40, 25.97)
Creatinine (µmol/L)	111.50 (81.00, 142.25)
eGFR (mL/min/1.73 m^2^) *	58.04 ± 26.50
Total Cholesterol (mg/dL)	154.68 (116.01, 193.35)
LDL-c (mg/dL)	77.34 (77.34, 116.01)
HDL-c (mg/dL)	38.67 (38.67, 38.67)
Triglycerides (mg/dL)	177.14 (88.57, 177.14)
ApoB (mg/mL)	86.00 (78.00, 103.50)
ANP (pg/mL)	0.43 (0.32, 0.60)
Ox-LDL (ng/mL) *	247.97 ± 67.50
Ox-LDL/ApoB (ratio)	2.70 (1.96, 3.48)
Ox-LDL/LDL-c (ratio)	98.64 (67.63, 124.93)
TNFα (pg/mL)	38.80 (32.01, 52.55)
IL8 (pg/mL)	13.98 (9.75, 18.45)
IL1b (pg/mL)	0.93 (0.60, 1.60)
IL6 (pg/mL)	24.72 (16.25, 32.93)
Insulin (pg/mL)	1615.57 (1180.50, 2386.68)
Leptin (pg/mL) *	8593.25 ± 3556.26
Adiponectin (ng/mL) *	141.87 ± 63.22
**Family History**	
Diabetes (%)	81.80
Hypertension (%)	25.50
CAD (%)	38.20
Cholesterol (%)	57.70
Stroke (%)	14.30
**Medication**	
Insulin (%)	47.37
HMG-CoA reductase inhibitors (%)	19.30
Metformin (%)	14.03
DPP4 inhibitors (%)	14.03
Sulfonylurea (%)	10.53
Calcium channel blockers (%)	8.77
ACE inhibitors (%)	7.02
NSAID (%)HMG-CoA reductase inhibitors (%)	7.02
Diuretics (%)	3.51
PPI (%)	3.51

Data are presented for the continuous variables as a mean (standard deviation, SD) or median (interquartile range, IQR) and as frequencies (percentages, %) for the categorical variables. *, Data are normally distributed. As control reference values, we used plasma from 13 randomly selected healthy volunteers with no recorded medical history to determine mean values for ANP (0.81 ± 0.59 pg/mL), ox-LDL (124 ± 19 ng/mL), TNFα (24.955 ± 7.941 pg/mL), IL-8 (18.256 ± 17.769 pg/mL), IL1b (2.464 ± 3.233), IL-6 (9.835 ± 4.821 pg/mL), insulin (1642.186 ± 1381.464 pg/mL), leptin (6125.617 ± 2719.795 pg/mL), and adipokine (225.398 ± 59.229 ng/mL). Abbreviations: BMI, body mass index; BP, blood pressure; HbA1c, hemoglobin HbA1c; LDL-c, low-density lipoprotein cholesterol; HDL-c, high-density lipoprotein cholesterol; Hs-CRP, high-sensitivity C-reactive protein; eGFR, estimated glomerular filtration rate; ApoB, apolipoprotein B; ANP, atrial natriuretic peptide; Ox-LDL, oxidized low-density lipoprotein; IL-8, interleukin 8; IL-1b, interleukin 1b; IL-6, interleukin 6; TNFα, tumor necrosis factor alpha; CAD, coronary artery disease; DPP4, dipeptidyl-peptidase 4; ACE, angiotensin-converting-enzyme; NSAID, non-steroidal anti-inflammatory drugs; PPI, proton-pump inhibitor.

**Table 2 ijms-26-01859-t002:** Matrix Spearman’s correlation between ANP, Ox-LDL, and other biochemical and clinical parameters in T2DM patients.

Variable		ANP	Age	Gender	SBP	DBP	A1c	Cre	Chol-t	LDL-c	HDL-c	TAG	BMI	Ox-LDL	Ox-LDL /ApoB	Ox-LDL /LDL-c
**ANP**	R	1.000	−0.100	0.041	0.072	−0.119	0.145	−0.013	0.128	0.091	−0.146	0.293	**0.297 ***	**−0.446 ****	**−0.423 ****	**−0.307 ***
	P	.	0.458	0.764	0.593	0.379	0.292	0.924	0.397	0.547	0.338	0.051	**0.025**	**0.001**	**0.001**	**0.038**
**Age**	R	−0.100	1.000	−0.076	−0.103	**−0.335 ***	0.044	0.186	−0.191	−0.241	0.042	−0.047	−0.063	0.082	0.119	0.245
	P	0.458	.	0.574	0.446	**0.011**	0.751	0.170	0.204	0.107	0.783	0.759	0.642	0.543	0.377	0.101
**Gender**	R	0.041	−0.076	1.000	0.042	−0.198	−0.023	**−0.429 ****	0.116	**0.360 ***	−0.060	−0.292	0.167	0.064	−0.009	−0.092
	P	0.764	0.574	.	0.758	**0.139**	0.865	**0.001**	0.443	**0.014**	0.697	0.051	0.214	0.635	0.950	0.544
**SBP**	R	0.072	−0.103	0.042	1.000	**0.549 *****	−0.055	0.106	0.216	**0.333 ***	−0.155	0.225	0.070	0.077	0.019	−0.228
	P	0.593	0.446	0.758	.	0.0001	0.692	0.436	0.150	**0.024**	0.310	0.138	0.607	0.571	0.889	0.128
**DBP**	R	−0.119	**−0.335 ***	−0.198	**0.549 *****	1.000	−0.087	−0.004	**0.317 ***	0.251	−0.262	0.048	−0.098	0.094	−0.009	−0.227
	P	0.379	**0.011**	0.139	**0.0001**	.	0.527	0.974	**0.032**	0.093	0.082	0.755	0.470	0.489	0.948	0.129
**A1c**	R	0.145	0.044	−0.023	−0.055	−0.087	1.000	−0.003	0.062	0.100	0.178	0.063	0.204	−0.098	**−0.318 ***	−0.105
	P	0.292	0.751	0.865	0.692	0.527	.	0.981	0.683	0.510	0.242	0.681	0.135	0.478	**0.018**	0.488
**Cre**	R	−0.013	0.186	**−0.429 ****	0.106	−0.004	−0.003	1.000	−0.089	−0.113	−0.134	0.091	−0.166	0.099	0.115	0.111
	P	0.924	0.170	**0.001**	0.436	0.974	0.981	.	0.555	0.456	0.382	0.551	0.221	0.466	0.400	0.463
**Chol-t**	R	0.128	−0.191	0.116	0.216	**0.317 ***	0.062	−0.089	1.000	**0.661 *****	−0.253	**0.297 ***	0.091	−0.048	**−0.327 ***	**−0.0562 *****
	P	0.397	0.204	0.443	0.150	**0.032**	0.683	0.555	.	**0.0001**	0.094	**0.047**	0.546	0.752	**0.026**	**0.0001**
**LDL-c**	R	0.091	−0.241	**0.360 ***	**0.333 ***	0.251	0.100	−0.113	**0.661 *****	1.000	−0.107	0.083	0.061	−0.032	**−0.409 ****	**−0.813 *****
	P	0.547	0.107	**0.014**	**0.024**	0.093	0.510	0.456	**0.0001**	.	0.484	0.588	0.685	0.833	**0.005**	**0.0001**
**HDL-c**	R	−0.146	0.042	−0.060	−0.155	−0.262	0.178	−0.134	−0.253	−0.107	1.000	0.064	−0.236	−0.079	−0.159	0.011
	P	0.338	0.783	0.697	0.310	0.082	0.242	0.382	0.094	0.484	.	0.682	0.119	0.607	0.296	0.942
**TAG**	R	0.293	−0.047	−0.292	0.225	0.048	0.063	0.091	**0.297 ***	0.083	0.064	1.000	0.109	−0.206	−0.227	−0.174
	P	0.051	0.759	0.051	0.138	0.755	0.681	0.551	**0.047**	0.588	0.682	.	0.477	0.176	0.134	0.253
**BMI**	R	**0.297 ***	−0.063	0.167	0.070	−0.098	0.204	−0.166	0.091	0.061	−0.236	0.109	1.000	−0.119	0.018	−0.054
	P	**0.025**	0.642	0.214	0.607	0.470	0.135	0.221	0.546	0.685	0.119	0.477	.	0.376	0.895	0.719
**Ox-LDL**	R	**−0.446 ****	0.082	0.064	0.077	0.094	−0.098	0.099	−0.048	−0.032	−0.079	−0.206	−0.119	1.000	**0.709 *****	**0.575 ****
	P	**0.001**	0.543	0.635	0.571	0.489	0.478	0.466	0.752	0.833	0.607	0.176	0.376	.	**0.0001**	**0.0001**
**Ox-LDL** **/ApoB**	R	**−0.423 *****	0.119	−0.009	0.019	−0.009	**−0.318 ***	0.115	**−0.327 ***	**−0.409 ****	−0.159	−0.227	0.018	**0.709 *****	1.000	**0.741 *****
	P	**0.001**	0.377	0.950	0.889	0.948	**0.018**	0.400	**0.026**	**0.005**	0.296	0.134	0.895	**0.0001**	.	**0.0001**
**Ox-LDL** **/LDLc**	R	**−0.307 ***	0.245	−0.092	−0.228	−0.227	−0.105	0.111	**−0.562 *****	**−0.813 *****	0.011	−0.174	−0.054	**0.575 *****	**0.741 *****	1.000
	P	**0.038**	0.101	0.544	0.128	0.129	0.488	0.463	**0.0001**	**0.0001**	0.942	0.253	0.719	**0.0001**	**0.0001**	.

Results are expressed as Spearman’s Rho coefficient (R) and probability value (*p*) for the 2-tailed Spearman’s correlation analysis. Significance: *, *p* ≤ 0.05, results are expressed as a Spearman’s Rho coefficient (R) and probability value (*p*) for the 2-tailed Spearman’s correlation analysis. Significance: *, *p* ≤ 0.05, **, *p* ≤ 0.001, and ***, *p* ≤ 0.0001. Abbreviations: ANP, atrial natriuretic peptide; SBP, systolic blood pressure; DBP, diastolic blood pressure; HbA1c, hemoglobin HbA1c; Cre, creatinine; Chol-t, total cholesterol; LDL-c, low-density lipoprotein cholesterol; HDL-c, high-density lipoprotein cholesterol; TAG, triglycerides; BMI, body mass index; Hs-CRP, high-sensitivity C-reactive protein; eGFR, estimated glomerular filtration rate; ApoB, apolipoprotein B; ANP, atrial natriuretic peptide; Ox-LDL, oxidized low-density lipoprotein.

**Table 3 ijms-26-01859-t003:** Spearman’s correlation matrix of ANP, Ox-LDL, and metabolic hormones and cytokines in T2DM patients.

		ANP	Ox-LDL	Adiponectin	Insulin	Leptin	IL8	IL1b	GLP1	IL6	TNFα
**ANP**	R	1.000	**−0.446 *****	**0.288 ***	0.004	−0.225	0.147	0.203	−0.048	**−0.305 ***	−0.213
	P	.	**0.0001**	**0.030**	0.976	0.093	0.275	0.130	0.725	**0.021**	0.112
**Ox-LDL**	R	**−0.446 *****	1.000	**−0.465 *****	−0.148	**0.353 ****	0.070	−0.136	0.225	**0.340 ****	**0.530 *****
	P	**0.0001**	.	**0.0001**	0.273	**0.007**	0.603	0.312	0.092	**0.010**	**0.0001**
**Adiponectin**	R	**0.288 ***	**−0.465 *****	1.000	0.047	−0.179	0.128	−0.109	−0.143	−0.068	**−0.276 ***
	P	**0.030**	**0.0001**	.	0.731	0.182	0.342	0.418	0.290	0.615	**0.037**
**Insulin**	R	0.004	−0.148	0.047	1.000	0.204	−0.072	−0.034	**0.316 ***	0.030	0.005
	P	0.976	0.273	0.731	.	0.129	0.592	0.804	**0.016**	0.824	0.969
**Leptin**	R	−0.225	**0.353 ****	−0.179	0.204	1.000	−0.181	−0.185	0.053	**0.353 ****	0.237
	P	0.093	**0.007**	0.182	0.129	.	0.178	0.168	0.698	**0.007**	0.075
**IL8**	R	0.147	0.070	0.128	−0.072	−0.181	1.000	**0.427 *****	−0.061	**0.265 ***	**0.364 ****
	P	0.275	0.603	0.342	0.592	0.178	.	**0.0001**	0.655	**0.046**	**0.005**
**IL1b**	R	0.203	−0.136	−0.109	−0.034	−0.185	**0.427 *****	1.000	−0.207	0.089	0.153
	P	0.130	0.312	0.418	0.804	0.168	**0.0001**	.	0.123	0.513	0.256
**IL6**	R	**−0.305 ***	**0.340 ****	−0.068	0.030	**0.353 ****	**0.265 ***	0.089	0.008	1.000	**0.529 *****
	P	**0.021**	**0.010**	0.615	0.824	**0.007**	**0.046**	0.513	0.954	.	**0.0001**
**TNFα**	R	−0.213	**0.530 *****	**−0.276 ***	0.005	0.237	**0.364 ****	0.153	0.114	**0.529 *****	1.000
	P	0.112	**0.0001**	**0.037**	0.969	0.075	**0.005**	0.256	0.398	**0.0001**	.

Results are expressed as a Spearman’s Rho coefficient (R) and probability value (*p*) for the 2-tailed Spearman’s correlation analysis. Significance: *, *p* ≤ 0.05, **, *p* ≤ 0.001, and ***, *p* ≤ 0.0001. Abbreviations: ANP, atrial natriuretic peptide; Ox-LDL, oxidized low-density lipoprotein; ADP, adiponectin; NGF, nerve growth factor; IL-8, interleukin 8; MCP1, monocyte chemoattractant protein-1; IL-1b, interleukin 1b; GLP1, glucagon-like peptide 1; IL-6, interleukin 6; INS, insulin; LEP, leptin; TNFα, tumor necrosis factor alpha; Hs-CRP, high-sensitivity C-reactive protein.

**Table 4 ijms-26-01859-t004:** Univariate regression analysis between Ox-LDL and ANP in T2DM patients.

Model (R^2^ = 0.144) **DV: Ox-LDL**	Unstandardized Coefficients		Standardized Coefficients	t	*p*	95% CI for B	
	B	SE	B			Lower Bound	Upper Bound
(Constant) **ANP**	302.641	19.850		15.247	0.000	262.862	342.420
	−117.231	38.613	−0.379	−3.036	**0.004**	−194.614	−39.849
Model (R^2^ = 0.101) **DV: Ox-LDL/ApoB**	Unstandardized Coefficients		Standardized Coefficients	t	*p*	95% CI for B	
	B	SE	B			Lower Bound	Upper Bound
(Constant) **ANP**	3.777	0.388		9.743	0.000	3.000	4.553
	−1.874	0.754	−0.318	−2.485	**0.016**	−3.385	−0.363
Model (R^2^ = 0.031) **DV: Ox-LDL/LDL-c**	Unstandardized Coefficients		Standardized Coefficients	t	*p*	95% CI for B	
	B	SE	B			Lower Bound	Upper Bound
(Constant) **ANP**	130.144	20.528		6.340	0.000	88.772	171.515
	−45.593	38.110	−0.177	−1.196	**0.238**	−122.399	31.213

*p*: probability value for the univariate regression analysis. *p* value ≤ 0.05 significant. CI: confidence of interval for beta (B) coefficient. Predictor ANP, independent variable (IV). Dependent variables (DV): ox-LDL, Ox-LD/ApoB, and Ox-LDL/LDL-c were individually entered in the regression model. R^2^ represents the proportion of variance in the dependent variable explained by the independent variable in the linear regression model.

**Table 5 ijms-26-01859-t005:** Multivariate regression analysis between ox-LDL, ANP, and clinical variables among patients with T2DM.

**A. Regression analysis examining the relationship between ANP and ox-LDL**								
Model 1 (R^2^ = 0.157) **DV: Ox-LDL**		Unstandardized Coefficients		Standardized Coefficients	t	*p*	95% CI for B	
		B	SE	B			Lower Bound	Upper Bound
**M1**	(Constant)	267.753	63.263		4.232	0.000	140.863	394.643
	**ANP**	−115.736	39.152	−0.374	−2.956	**0.005**	−194.265	−37.207
	Age	0.714	0.854	0.106	0.836	0.407	−0.999	2.427
	Gender	−6.429	16.970	−0.048	−0.379	0.706	−40.466	27.609
Model (R^2^ = 0.145) **DV: Ox-LDL**		Unstandardized Coefficients		Standardized Coefficients	t	*p*	95% CI for B	
		B	SE	B			Lower Bound	Upper Bound
**M2**	(Constant)	267.155	73.138		3.653	0.001	120.253	414.058
	**ANP**	−105.978	41.248	−0.358	−2.569	**0.013**	−188.827	−23.128
	Age	0.689	0.852	0.106	0.810	0.422	−1.021	2.400
	Gender	−4.691	16.916	−0.036	−0.277	0.783	−38.668	29.286
	A1c	−0.680	5.658	−0.017	−0.120	0.905	−12.044	10.684
Model (R^2^ = 0.378) **DV: Ox-LDL**		Unstandardized Coefficients		Standardized Coefficients	t	*p*	95% CI for B	
		B	SE	B			Lower Bound	Upper Bound
**M3**	(Constant)	168.283	118.323		1.422	0.167	−75.408	411.975
	**ANP**	−81.813	54.394	−0.289	−1.504	**0.145**	−193.840	30.214
	Age	2.163	1.253	0.299	1.726	0.097	−0.418	4.743
	Gender	9.318	24.018	0.071	0.388	0.701	−40.147	58.783
	A1c	5.238	7.962	0.140	0.658	0.517	−11.160	21.637
	Chol-t	10.726	16.700	0.226	0.642	0.527	−23.667	45.120
	LDL-c	1.163	19.339	0.018	0.060	0.953	−38.667	40.993
	HDL-c	−15.016	32.044	−0.118	−0.469	0.643	−81.011	50.980
	Triglycerides	−12.931	13.199	−0.194	−0.980	0.337	−40.116	14.254
	Hs-CRP	−0.568	0.345	−0.290	−1.649	0.112	−1.278	0.142
	BMI	−2.146	1.914	−0.212	−1.121	0.273	−6.088	1.797
**B. Regression analysis examining the relationship between ANP and ox-LDL/ApoB ratio**								
Model 1 (R^2^ = 0.130) **DV: Ox-LDL/ApoB**		Unstandardized Coefficients		Standardized Coefficients	t	*p*	95% CI for B	
		B	SE	B			Lower Bound	Upper Bound
**M1**	(Constant)	2.864	1.226		2.337	0.023	0.406	5.322
	**ANP**	−1.836	0.758	−0.311	−2.420	**0.019**	−3.357	−0.314
	Age	0.019	0.017	0.149	1.162	0.250	−0.014	0.052
	Gender	−0.191	0.329	−0.074	−0.580	0.564	−0.850	0.469
Model 2 (R^2^ = 0.169) **DV: Ox-LDL/ApoB**		Unstandardized Coefficients		Standardized Coefficients	t	*p*	95% CI for B	
		B	SE	B			Lower Bound	Upper Bound
**M2**	(Constant)	4.226	1.440		2.935	0.005	1.334	7.119
	**ANP**	−1.289	0.812	−0.218	−1.587	**0.119**	−2.920	0.343
	Age	0.021	0.017	0.160	1.232	0.224	−0.013	0.054
	Gender	−0.187	0.333	−0.073	−0.562	0.576	−0.856	0.482
	A1c	−0.199	0.111	−0.247	−1.790	0.079	−0.423	0.024
Model 3 (R^2^ = 0.373) **DV: Ox-LDL/ApoB**		Unstandardized Coefficients		Standardized Coefficients	t	*p*	95% CI for B	
		B	SE	B			Lower Bound	Upper Bound
**M3**	(Constant)	3.513	2.425		1.449	0.160	−1.481	8.507
	**ANP**	−1.365	1.115	−0.236	−1.225	**0.232**	−3.661	0.930
	Age	0.029	0.026	0.200	1.148	0.262	−0.023	0.082
	Gender	−0.238	0.492	−0.088	−0.483	0.633	−1.251	0.776
	A1c	−0.074	0.163	−0.097	−0.453	0.654	−0.410	0.262
	Chol-t	0.349	0.342	0.360	1.020	0.318	−0.356	1.054
	LDL-c	−0.724	0.396	−0.558	−1.826	0.080	−1.540	0.093
	HDL-c	0.313	0.657	0.120	0.476	0.638	−1.040	1.665
	Triglycerides	−0.467	0.270	−0.342	−1.727	0.097	−1.024	0.090
	Hs-CRP	−0.011	0.007	−0.266	−1.507	0.144	−0.025	0.004
	BMI	0.010	0.039	0.048	0.250	0.805	−0.071	0.091
**C. Regression analysis examining the relationship between ANP and ox-LDL/LDL-c ratio**								
Model 1 (R^2^ = 0.159) **DV: Ox-LDL/LDL-c**		Unstandardized Coefficients		Standardized Coefficients	t	*p*	95% CI for B	
		B	SE	B			Lower Bound	Upper Bound
**M1**	(Constant)	35.967	68.264		0.527	0.601	−101.797	173.730
	**ANP**	−43.459	36.777	−0.169	−1.182	**0.244**	−117.678	30.761
	Age	1.941	0.894	0.311	2.171	0.036	0.137	3.745
	Gender	−16.543	17.044	−0.140	−0.971	0.337	−50.938	17.853
Model 2 (R^2^ = 0.182) **DV: Ox-LDL/LDL-c**		Unstandardized Coefficients		Standardized Coefficients	t	*p*	95% CI for B	
		B	SE	B			Lower Bound	Upper Bound
**M2**	(Constant)	72.050	75.894		0.949	0.348	−81.221	225.322
	**ANP**	−28.585	39.208	−0.111	−0.729	**0.470**	−107.767	50.597
	Age	2.076	0.901	0.333	2.304	0.026	0.256	3.895
	Gender	−16.753	17.012	−0.142	−0.985	0.330	−51.109	17.602
	A1c	−5.869	5.439	−0.164	−1.079	0.287	−16.852	5.114
Model 3 (R^2^ = 0.271) **DV: Ox-LDL/LDL-c**		Unstandardized Coefficients		Standardized Coefficients	t	*p*	95% CI for B	
		B	SE	B			Lower Bound	Upper Bound
**M3**	(Constant)	90.788	101.443		0.895	0.379	−118.139	299.714
	**ANP**	−50.042	46.634	−0.181	−1.073	**0.293**	−146.087	46.004
	Age	2.373	1.074	0.336	2.209	0.037	0.160	4.585
	Gender	−22.261	20.591	−0.173	−1.081	0.290	−64.669	20.148
	A1c	−0.620	6.826	−0.017	−0.091	0.928	−14.680	13.439
	Chol-t	19.187	14.317	0.414	1.340	0.192	−10.300	48.674
	LDL-c	−48.789	16.581	−0.787	−2.943	0.007	−82.938	−14.641
	HDL-c	24.685	27.473	0.198	0.899	0.377	−31.895	81.266
	Triglycerides	−17.183	11.316	−0.263	−1.518	0.141	−40.490	6.123
	Hs-CRP	−0.518	0.296	−0.271	−1.751	0.092	−1.126	0.091
	BMI	−0.170	1.641	−0.017	−0.104	0.918	−3.551	3.210

*p*: probability value for each independent variable entered into the regression model (M1–M3). *p* value ≤ 0.05 significant. CI: confidence of interval for beta (B) coefficient. Predictors (independent variables): ANP and other variables such as age, gender, A1C, total cholesterol, LDL-c, HDL-c, triglycerides, hs-CRP, and BMI. Dependent variable (DV): ox-LDL, Ox-LD/ApoB, or Ox-LDL/LDL-c. R^2^: proportion of variance in the dependent variable explained by the independent variable in the linear regression model.

**Table 6 ijms-26-01859-t006:** Regression analysis of potential predictors among plasma adipokines and cytokines for ox-LDL levels in patients with T2DM.

**A. Multi-regression outcome analysis**								
**Model**		Unstandardized Coefficients		Standardized Coefficients	t	*p*	95% CI for B	
		B	SE	B			Lower Bound	Upper Bound
**Model** **(R^2^ = 0.612)**	(Constant)	236.369	63.234		3.738	0.001	108.845	363.892
	**ANP**	−72.823	32.436	−0.246	−2.245	**0.030**	−138.236	−7.410
	Age	0.481	0.652	0.074	0.739	0.464	−0.833	1.795
	Gender	−12.108	13.501	−0.094	−0.897	0.375	−39.336	15.120
	HbA1c	1.964	4.408	0.048	0.446	0.658	−6.925	10.853
	IL-8	−0.024	0.872	−0.004	−0.028	0.978	−1.782	1.734
	IL-1b	−4.742	4.700	−0.112	−1.009	0.319	−14.221	4.737
	IL-6	−1.038	0.695	−0.191	−1.493	0.143	−2.440	0.364
	TNFα	1.630	0.523	0.401	3.116	0.003	0.575	2.685
	Adiponectin	−0.382	0.117	−0.364	−3.273	0.002	−0.618	−0.147
	Insulin	−0.008	0.007	−0.114	−1.104	0.276	−0.022	0.006
	Leptin	0.006	0.002	0.310	2.684	0.010	0.001	0.010
**B. Stepwise Outcome Analysis**								
**Models**		Unstandardized Coefficients		Standardized Coefficients	t	*p*	95% CI for B	
		B	SE	B			Lower Bound	Upper Bound
**M1** **(R^2^ = 0.277)**	(Constant)	326.238	19.326		16.881	0.0000	287.476	365.000
	Adiponectin	−0.552	0.123	−0.526	−4.501	0.0001	−0.798	−0.306
**M2** **(R^2^ = 0.433)**	(Constant)	243.324	27.864		8.733	0.0000	187.411	299.237
	Adiponectin	−0.462	0.112	−0.440	−4.118	0.0001	−0.687	−0.237
	TNFα	1.645	0.434	0.405	3.792	0.0004	0.774	2.515
**M3** **(R^2^ = 0.501)**	(Constant)	202.174	30.647		6.597	0.0000	140.648	263.700
	Adiponectin	−0.419	0.107	−0.399	−3.897	0.0003	−0.635	−0.203
	TNFα	1.411	0.420	0.348	3.358	0.0010	0.568	2.255
	Leptin	0.005	0.002	0.272	2.641	0.0110	0.001	−0.009
**M4** **(R^2^ = 0.546)**	(Constant)	235.398	33.115		7.109	0.0000	168.885	301.912
	Adiponectin	−0.387	0.105	−0.369	−3.705	0.0005	−0.597	−0.177
	TNFα	1.391	0.405	0.343	3.433	0.0012	0.577	2.205
	Leptin	0.004	0.002	0.233	2.313	0.0487	0.001	0.008
	ANP	−64.664	29.155	−0.218	−2.218	0.0311	−123.224	−6.104

Multi-regression was conducted to examine the effects of adipokines and cytokines on the association of ANP with ox-LDL. *p* value ≤ 0.05 was considered significant. CI: confidence of interval for beta (B) coefficient. Predictors (independent variables) such as ANP, age, gender, HbA1c, Il-8, IL-1b, IL-6, TNFα, adiponectin, insulin, and leptin were included in the model. Stepwise linear regression analysis was conducted to identify the strongest predictor associated with ox-LDL (dependent variable). Independent predictors (ANP, age, gender, HbA1c, IL8, IL1b, IL6, TNFα, adiponectin, insulin, and leptin) were included in the models (M1–M3). The criteria for entering variables into the models during stepwise regression were set at F ≤ 0.050, while the criteria for removal were F ≥ 0.100. R^2^ represents the proportion of variance in the dependent variable explained by the independent variables in the linear regression model (M1–M3).

**Table 7 ijms-26-01859-t007:** Regression analysis of alternative mediation effects of circulating adipokines and cytokines on the relationship between plasma ANP and ox-LDL in patients with T2DM.

Mediator	Effect (β)	SE	t	*p*	LLCI	ULCI	Mediation (%)
Type							
**Adiponectin**							
Total (c)	−117.231	38.613	−3.036	0.004	−194.614	−39.848	24.36
Direct (c′)	−88.672	35.038	−2.531	0.014	−158.921	−18.424	
Indirect (a × b)	−28.559	19.320	-	NS	−75.234	0.645	
**Leptin**							
Total (c)	−117.231	38.613	−3.036	0.004	−194.614	−39.848	10.04
Direct (c′)	−105.470	38.076	−2.770	0.008	−181.807	−29.132	
Indirect (a × b)	−11.762	20.037	-	NS	−69.319	5.728	
**Insulin**							
Total (c)	−117.231	38.613	−3.036	0.004	−194.614	−39.848	0.13
Direct (c′)	−117.072	38.725	−3.019	0.004	−194.827	−39.317	
Indirect (a × b)	−0.159	13.657	-	NS	−16.684	42.090	
**TNFα**							
Total (c)	−117.231	38.613	−3.036	0.004	−194.614	−39.848	14.01
Direct (c′)	−100.803	33.680	−2.993	0.004	−168.331	−33.275	
Indirect (a × b)	−16.428	17.542	-	NS	−59.327	10.515	
**IL-6**							
Total (c)	−117.231	38.613	−3.036	0.004	−194.614	−39.848	13.79
Direct (c′)	−101.060	38.883	−2.599	0.012	−179.017	−23.103	
Indirect (a × b)	−16.171	14.352	-	NS	−50.777	3.201	
**IL-1b**							
Total (c)	−117.231	38.613	−3.036	0.004	−194.614	−39.848	1.80
Direct (c’)	−115.114	38.867	−2.962	0.005	−193.038	−37.189	
Indirect (a × b)	−2.117	6.907	-	NS	−11.300	18.458	
**IL-8**							
Total (c)	−117.231	38.613	−3.036	0.004	−194.614	−39.848	4.88
Direct (c′)	−122.956	39.950	−3.078	0.003	−203.051	−42.861	
Indirect (a × b)	−5.725	9.155	-	NS	−7.799	30.573	

Data are shown as the effect of mediating variables on the relationship between ANP (independent variable) and ox-LDL (dependent variable). Hayes Process Macro (V4.2) was used to generate results. In a simple mediation (model 4), the indirect effect (ab) constitutes the extent to which the independent variable (ANP) influences the dependent variable (ox-LDL) through the individual mediator. The total mediation effect (c) equals the sum of the model’s direct and indirect mediation effects (c′ + ab). LLCI, lower-level confidence interval, ULCI, upper-level confidence interval. *p* value ≤ 0.05 is significant. NS: not significant.

## Data Availability

Data are contained within the article.

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
