# Peer review of "Plasma Atrial Natriuretic Peptide Predicts Oxidized Low-Density Lipoprotein Levels in Type 2 Diabetes Mellitus Patients Independent of Circulating Adipokine and Cytokine"

_ijms, 2025, doi:10.3390/ijms26051859_

Round 1
Reviewer 1 Report
Comments and Suggestions for Authors
The authors have investigated the association between ANP and ox-LDL levels in obese diabetic patients- They found that patients with T2DM had elevated circulating ox-LDL levels and a slight, though not statistically significant, decrease in plasma ANP levels compared to healthy controls.
Comments
1. Clinical characteristics of the control group (e.g., age, sex, BMI, etc.) have been included in Table 1 for comparison with the obese diabetic patients."
2. Was there a significant difference in the levels of adiponectin, leptin, and TNF-alpha between obese diabetic patients and the control group?
3. Why do you measure Ox-LDL/ApoB and Ox-LDL/LDL-c ? Is any difference in the control group?
4. Which are the values of sensibility and specificity of ANP and oxLDL assay?
Reviewer 2 Report
Comments and Suggestions for Authors
In the manuscript “Plasma Atrial Natriuretic Peptide Predicts Oxidized Low-Density Lipoprotein Levels in Type-2 Diabetes Mellitus Patients Independent of Adipokine and Cytokine Fluctuations” the authors investigated the possible impact of ANP and ox-LDL in T2DM patients. Although the authors have taken great care to assess the data using various parameters and various statistical approaches, there are some fundamental concerns regarding the assessment and interpretation of the data.
As the study was cross-sectional, one cannot refer to ANP levels as “predicting” ox-LDL, as this implies more than one time-point/ changes over time or reactivity testing. Adjusting for adipokine and cytokine levels do not equate to fluctuations, as these levels were not assessed during a metabolic stressor which would induce such fluctuations (only fasting samples were measured).
The authors are commended for approaching various possible angles by including markers such as adipokines, NGF, leptin etc – however, the mechanistic inferences of such inclusions become unclear, as too many variables are introduced in too small a sample size (N=57 T2DM patients vs 17 stated healthy controls, yet no data in the basic comparisons are resported). It is also kindly advised that these assessments are repeated in a normoglycemic control group, to assess whether these associations are exclusive in T2DM (as many reflect physiological compensatory mechanisms when energy metabolism is compromised). It may be considered to employ age, sex (and, only if possible BMI) matched controls.
There is not sufficient motivation to include pertinent variables such as NGF, GLP, adiponectin and leptin, within the current aims and objectives (please see a possible focus suggestion below).
Thus the researchers are kindly requested to simplify the approach, adjust the interpretation to allow focus on such an important pathological mechanism. As a kind suggestion and an attempt to assist in this reasoning, a possible approach and discussion (to focus on the main parameters in the title) to consider may be:
Importantly, the relationship between atrial natriuretic peptide (ANP) and oxidized low-density lipoprotein (Ox-LDL) is significant in the context of cardiovascular health, particularly concerning inflammation, lipid metabolism, and vascular function. This relationship is further complicated in individuals with T2DM, where metabolic dysregulation can exacerbate cardiovascular risks – this is a well-established link (and sources have been provided as links to the authors for their consideration and convenience).
Recent studies have highlighted ANP's role in lipid metabolism, where it can inhibit the production of inflammatory cytokines and improve insulin sensitivity in adipose tissue (https://www.mdpi.com/1422-0067/20/13/3265; https://academic.oup.com/eurheartj/article-abstract/35/7/419/442216?redirectedFrom=fulltext&login=false). Ox-LDL can stimulate the migration of vascular smooth muscle cells (VSMCs), which is a critical process in plaque formation (https://pubmed.ncbi.nlm.nih.gov/9314840/) thus atherosclerosis risk etc.
Some aspects to consider when considering the relationship between ANP and ox-LDL (just to aid the researchers in simplifying and focussing the analyses and discussion)
- Inhibition of Inflammation: ANP has been shown to counteract the pro-inflammatory effects of Ox-LDL. It can inhibit the migration of VSMCs stimulated by Ox-LDL, thus potentially reducing the progression of atherosclerosis (https://pubmed.ncbi.nlm.nih.gov/9314840/ ).
- Lipid Metabolism Regulation: ANP influences lipid metabolism by promoting lipolysis and enhancing lipid oxidation. This action can counteract some adverse effects associated with elevated levels of Ox-LDL (https://pmc.ncbi.nlm.nih.gov/articles/PMC8355588/)
- Endothelial Function: ANP may help maintain endothelial function in the presence of Ox-LDL by promoting vasodilation and reducing oxidative stress, which is often heightened by oxidized lipoproteins (https://pmc.ncbi.nlm.nih.gov/articles/PMC8355588/)
Why in T2DM (the researchers population of interest, which should be interpreted versus healthy controls)
In individuals with T2DM, the dynamics between ANP and Ox-LDL are altered due to several factors that the authors can consider:
- Increased Oxidative Stress: Elevated oxidative stress and inflammation, which can lead to increased levels of Ox-LDL. This environment may overwhelm the protective effects of ANP (again requires comparisons with a healthy control group)
- Dysregulated Lipid Metabolism: Dyslipidemia, characterized by elevated triglycerides and altered LDL profiles. The presence of high levels of Ox-LDL may impair the beneficial actions of ANP on lipid metabolism (a possible aspect to consider should the association prove significant, and if it can be compared to healthy controls)
- Impaired ANP Response: Research suggests that insulin resistance can affect the secretion and action of natriuretic peptides. In T2DM, lower responsiveness to ANP may reduce its protective cardiovascular effects against the detrimental impacts of Ox-LDL…
These are just some aspects to consider to aid in streamlining the approach, and still highlight the novelty of any possible interaction, from a mechanistic perspective. The study holds promise, yet requires significant refinement.
Comments on the Quality of English LanguageN/A
